# Remote Eradication of Bacteria on Orthopedic Implants via Delayed Delivery of Polycaprolactone Stabilized Polyvinylpyrrolidone Iodine

**DOI:** 10.3390/jfb13040195

**Published:** 2022-10-19

**Authors:** Yikai Wang, Wangsiyuan Teng, Zengjie Zhang, Siyuan Ma, Zhihui Jin, Xingzhi Zhou, Yuxiao Ye, Chongda Zhang, Zhongru Gou, Xiaohua Yu, Zhaoming Ye, Yijun Ren

**Affiliations:** 1Department of Orthopedics, Renmin Hospital of Wuhan University, Jiefang Road 238, Wuhan 430060, China; 2Department of Orthopedics, Centre for Orthopaedic Research, Orthopedics Research Institute of Zhejiang University, The Second Affiliated Hospital, Zhejiang University School of Medicine, Hangzhou 310009, China; 3Key Laboratory of Motor System Disease Research and Precision Therapy of Zhejiang Province, Hangzhou 310009, China; 4School of Material Science and Engineering, University of New South Wales, Sydney 2052, Australia; 5New York University Medical Center, New York University, New York, NY 10016, USA; 6Bio-Nanomaterials and Regenerative Medicine Research Division, Zhejiang-California International Nanosystem Institute, Zhejiang University, Hangzhou 310058, China

**Keywords:** antibacterial coating, micro-arc oxidation, orthopedic implants, polycaprolactone, povidone-iodine

## Abstract

Bacteria-associated late infection of the orthopedic devices would further lead to the failure of the implantation. However, present ordinary antimicrobial strategies usually deal with early infection but fail to combat the late infection of the implants due to the burst release of the antibiotics. Thus, to fabricate long-term antimicrobial (early antibacterial, late antibacterial) orthopedic implants is essential to address this issue. Herein, we developed a sophisticated MAO-I_2_-PCLx coating system incorporating an underlying iodine layer and an upper layer of polycaprolactone (PCL)-controlled coating, which could effectively eradicate the late bacterial infection throughout the implantation. Firstly, micro-arc oxidation was used to form a microarray tubular structure on the surface of the implants, laying the foundation for iodine loading and PCL bonding. Secondly, electrophoresis was applied to load iodine in the tubular structure as an efficient bactericidal agent. Finally, the surface-bonded PCL coating acts as a controller to regulate the release of iodine. The hybrid coatings displayed great stability and control release capacity. Excellent antibacterial ability was validated at 30 days post-implantation via in vitro experiments and in vivo rat osteomyelitis model. Expectedly, it can become a promising bench-to-bedside strategy for current infection challenges in the orthopedic field.

## 1. Introduction

Infections are a devastating complication after implantation, which can greatly increase the time to cure the disease and place a huge financial burden on the patient and the healthcare system [1,2,3]. Based on the prediction by Kurtz et al., the annual expense of periprosthetic joint infection in the medical system in the U.S. may exceed $1.62 billion in 2020 [4]. Indeed, prosthetic joint infection (PJI), especially osteomyelitis, is the most common reason for revision surgery (~74%) [5]. For some severe cases of prosthetic infection, patients have to undergo removal of all components, aggressive debridement, second revision with variable antibiotics treatment, and even amputation sometimes [6,7]. Prosthetic infections could originate from bacteria contamination at the surgical site during surgery and blood-borne microorganism in patients with poor immunity post-operation. Various microorganisms can rapidly adhere to the implants and form a dense biofilm to protect them from host immunity and antibiotic lethality [8]. Furthermore, the diagnosis of PJI, including identifying the species of pathogenic bacteria and their antibacterial sensitivity, can be difficult due to their invisibility and inaccessibility, thus making PJI problematic to detect and treat in time [9]. Accordingly, it is urgent to develop a new method that can prevent these destructive infections under feasible conditions.

Over the past three decades, multiple drug-device combination systems have been attempted to prevent the infection, such as antibiotic-loaded implants [10], antibiotic-impregnated polymeric bone cement [11], and metal ions-loaded coating system (e.g., Ag^+^, Cu^2+^, and Zn^2+^, etc.) [12,13,14], which has achieved great success in fighting pathogenic microorganism [15]. Although several strategies have been proposed in the literature, there is no consensus on the best treatment to control peri-implant infections [16,17,18]. What is more, these treatments even present many shortcomings in antibacterial application [2,3,19]. The vast majority of antimicrobial strategies lack a long-acting antimicrobial effect and are only short-term antimicrobial applications. For example, many antibiotics-eluting implants suffer from “burst release” when implanted in the body, which are only effective in the early stage, with no antibacterial activities in the late stage [20,21]. Additionally, sporadic attempts at long-acting antimicrobial strategies have also been reported, but unfortunately, unstable properties, less than one-month antimicrobial aging, and poor biocompatibility limit their application. In addition, for antibiotics application, they are gradually restricted owing to bacterial resistance, tolerance, and/or persistence as the emergence of multi-drug resistant bacteria in the field of orthopedics [22,23]. For antibacterial metal ions, they have also exposed many drawbacks, such as poor biocompatibility and intrinsic toxicity [24,25], less effective bone repair, retarded bone remodeling, and more painful patients’ recovery in clinical applications [26]. Hence, it is crucial to develop a kind of antibacterial strategy, which has a long-term antimicrobial effect, good biocompatibility, and no drug resistance in orthopedics.

Titanium alloys are widely used orthopedic implants of which surface modification has been tried extensively in antimicrobial applications [27]. The microstructure modification could derive conveniently from titanium alloy by combining acidic solution reactions, such as acid etching, anodization, and micro-arc oxidization (MAO) [28,29]. Attempts to micro-arc oxidize the titanium alloy and load versatile bactericide for antibacterial application have been reported many times and have achieved certain antibacterial effects [30,31]. However, unfortunately, the above issues of short antibacterial aging, poor biocompatibility, and poor stability have not been solved. As a complex of polyvinylpyrrolidone (PVP) and iodine, PVP-I_2_ has attracted more and more attention from scientists because of its unique bactericidal features in the past 20 years [32,33]. These excellent features include high antibacterial properties; a broad-spectrum antibacterial effect against bacteria, molds, and some viruses; persistence; outstanding biocompatibility; and non-irritation [34,35,36]. Povidone-iodine has been widely used in medical fields, such as disinfection of surgical sites for hundreds of years, but it is incredible that no bacterial resistance has been detected so far [35]. Thus, the combination of iodine and medical implants, especially titanium alloy may be a promising discovery in the antibacterial field. However, iodine is extremely unstable and will sublime very quickly at room temperature, which makes it very difficult to be loaded on the surface of titanium alloy [34]. A simple system combining iodine and titanium alloy is inevitably limited in poor control of release kinetics resulting in the primarily burst iodine release, indicating ineffective antibacterial applications. Thus, a special controlled surface structure and release controller coating on titanium alloy to stablize the iodine is needed [36]. Specifically, we highlight a hierarchical titania/iodine (TiO_2_/I_2_) coating for titanium alloy implants where PVP-I_2_ is loaded in the micro-structured TiO_2_ pipes achieved by micro-arc oxidization. Polycaprolactone (PCL), one of the most popular synthetic polymers, was widely applied in drug delivery and tissue engineering owing to its good biocompatibility, biodegradability, and excellent mechanical properties [37]. The specific topology of the titanium alloy surface provides effective conditions for the adhesion and stability of PCL, which can significantly increase the bonding strength between the soft and hard materials and form a coating on the surface of the titanium alloy more uniformly. Therefore, we hypothesized integration of PCL, PVP-I_2_, and micro-structured TiO_2_ could serve as an effective control-released anti-infection system and overcome the disadvantages of current drug-eluting implants.

Herein, we developed a PVP-I_2_-integrated PCL-sealed coating on the surface of titanium alloy via micro-arc oxidation (MAO), electrophoresis, and dip-coating technique. This sophisticated MAO-I_2_-PCLx coating system exerts a long-lasting antimicrobial performance and can efficiently remove bacteria both at the early and late stages after implantation. Specially, the PCL coating is designed to control the release of iodine and balance the antibacterial performance. We demonstrated that iodine can be stabilized to release for at least four weeks controlled by PCL coating, while it can be only maintained for one day during release from a simple titanium alloy/PVP-I_2_ coating. In addition, the ex vivo and in vivo results confirmed that the PCL coating with the PVP-I_2_ antibacterial agent possesses remarkable antibacterial ability against *Staphylococcus aureus* (*S. aureus*) and *Escherichia coli* (*E. coli*) as well as excellent biocompatibility properties even in the challenging infected scenario.

## 2. Materials and Methods

### 2.1. Materials

Titanium alloy substrates (10 mm × 10 mm × 1 mm) and rods (1 mm of diameter and 20 mm of length) were obtained from the Institute of Orthopedics, Zhejiang University (Hangzhou, China). Sulfuric acid (H_2_SO_4_), phosphoric acid (H_3_PO_4_), hydrogen peroxide (H_2_O_2_), potassium iodide (KI), povidone-iodine (PVPI), and PCL particles (CAS number: 24980-41-4, molecular weight 80,000) were provided by Aladdin Industrial Company (Shanghai, China).

### 2.2. Preparation of Main Experimental Reagents

#### 2.2.1. Preparation of Anodic Oxidation Electrolytic Solution

The acid electrolyte solution was prepared with H_2_SO_4_ (35 g/L), H_3_PO_4_ (25 g/L), and H_2_O_2_ (10 g/L) solution, which is shown in Appendix A.

#### 2.2.2. Preparation of KI/PVPI Mixed Electrophoresis Solution

The electrophoresis solution was prepared by the 1 wt% KI solution, 1 wt% PVPI, and distilled H_2_O solution, which is shown in Appendix A.

#### 2.2.3. Preparation of PCL Gradient Solution

The PCL gradient solution (low, medium, and high concentration) were prepared by adding gradient weight PCL particles (10 g, 75 g, and 150 g) to 400 mL dichloromethane, which was magnetically stirred for 20–30 min until the solid particles were completely dissolved. The solution has to be prepared and sealed for immediate use.

### 2.3. The Preparation of the MAO-I_2_-PCLx Coatings (x = 1, 2, 3)

#### 2.3.1. Samples Preparation

After polishing with sandpaper with gradient roughness (No. 600, 800, 1000, 2000), sandblasting treatment, and ultrasonic cleaning by acetone (≥99.5%), ethanol (≥99.5%), and distilled water (each for 10 min), the titanium alloy was treated into the mix solution containing hydrogen fluoride (HF) (≥40%, 20 mL/L), nitric acid (HNO_3_) (65.0–68.0%, 80 mL/L), and phosphoric acid (H_3_PO_4_) (85%, 15 mL/L) for pickling for 15 min. Then, the substrates were treated with ultrasonic cleaning 3 times (5 min for each time), followed by nitrogen blow dry.

#### 2.3.2. Micro Arc Oxidation Treatment

The pretreated substrates were connected to the anode of the anodic oxidation cell, and the platinum sheet was connected to the cathode, which was separated by 15 cm in parallel. MAO was conducted using an anodized oxidation tank in the above-mentioned acidic anodizing solution with a pulsing power supply (JHMAO-6H; Environmental Golden-Arc, Beijing, China) at 150 V for 5 min at 30 °C. The frequency and duty cycle were set to 1200 H_Z_ and 10%, respectively. In the process of oxidation, spark sputtering can be seen on the surface of the titanium alloy. After the reaction, the substrates were placed into the aqueous solution for ultrasonic immersion 3 times (5 min for each time) and then dried for subsequent testing.

#### 2.3.3. The Process of Electrophoresis and Iodine Loading

The treated substrates were connected to the anode of the electrophoresis tank and a pure platinum plate was connected to the cathode, which was separated by 20 cm between the anode and cathode plate in the above-mentioned mixed electrophoresis solution of PVPI and KI. Then, it was reacted for 30 min under the condition of a constant voltage of 30 V and below 20 °C to load the iodine. Then, the substrates are rinsed by ultrapure water 3 times, each time for 3 min. Finally, the substrates were dried by cold nitrogen for subsequent experiments.

#### 2.3.4. Construction of the PCL Controlled Release Coating

The iodine-treated sample was immersed in the polycaprolactone solution (low, medium, and high concentration) for 1 min. Then, the sample was placed in a fume hood for 12 h, and finally sterilized by irradiation and placed in the refrigerator at 4 °C for subsequent experiments.

### 2.4. The Characterization of the MAO-I_2_-PCLx Coatings(x = 1, 2, 3)

An ordinary camera was used to observe the changes in the color and morphology of the surface of different substrates. A field emission scanning electron microscope and an energy spectrometer (SEM&EDS, Hitachi SU-8010/Gemini SEM 300, Hitachi, Tokyo, Japan) were used to observe the surface morphology, surface element content, and distribution of the material. An x-ray photoelectron spectrometer (XPS; Thermo Scientific K-Alpha, New York, NY, USA) was used to determine the surface elements and chemical composition. The surface roughness of the sample was measured by 3D atomic force microscope (AFM) images (NT-MDT Prima; Bruker Dimension Edge; Bruker Dimension ICON, Bruker, Berlin, Germany).

### 2.5. Bacterial Species

*S. aureus* (25923) and *E. coli* (8099) were all derived from ATCC and confirmed by bacterial identification.

### 2.6. Iodine Ion Release Experiment and Determination of Iodine Content on the Surface of Titanium Alloy over Time

In 20 mL Tris buffer solution (pH = 7.4), all substrates were immersed in it to simulate the physiological environment of the human body. After soaking at different time intervals (12 h, 1, 3, 7, 15, 21, and 30 days), the 1.0 mL supernatant was centrifuged, and inductively coupled plasma atomic emission spectrometry (ICP-AES, IRIS Advantage ER/S, Thermo Fisher Scientific, New York, NY, USA) analyzed the concentration of iodide ions released at different time points. At each time point, 1 mL of the same fresh Tris buffer was added to the buffer system of the original test sample to keep the solution volume constant. Three sets of parallel sample experiments were used to determine the cumulative release curve. A field emission scanning electron microscope energy spectrometer (SEM&EDS, Hitachi SU-8010/Gemini SEM 300) was used to measure the iodine content on the surface of the material at different time points (immediately, 3 days, 15 days, and 30 days) after sample preparation.

### 2.7. Biocompatibility Test

#### 2.7.1. Cell Culture

SD rat femur bone marrow mesenchymal stem cells (BMSCs) were used for in vitro cell compatibility tests. BMSCs were cultured in a low-sugar DMEM medium containing 10% fetal bovine serum, 100 μg/mL streptomycin, and 100 ug/mL penicillin at 37 °C and 5% CO_2_. The cell culture medium was changed every two days. All experiments used the third-generation BMSCs.

#### 2.7.2. CCK-8 Experiment

The CCK-8 experiment was used to determine the cytotoxicity of various samples. A total of 1 mL of BMSCs cell suspension (2 × 10^4^ cells/mL) was added to different substrates and cultured for 1, 3, and 5 days, respectively. At different time points, 220 μL of the mixture solution was freshly cultured without serum base (200 μL), and CCK-8 reagent (20 μL) (*v/v* = 10:1) was added to the well plate of each substrate and incubated at 37 °C for 2 h. Finally, the OD value of the supernatant was measured by spectrophotometry at 450 nm to determine the cell proliferation activities.

#### 2.7.3. Cytoskeleton Staining

Bone marrow mesenchymal stem cells (BMSCs) were cultured on the surface of each group of substrates for 3 days in the low-sugar DMEM medium containing 10% fetal bovine serum, 100 μg/mL streptomycin, and 100 ug/mL penicillin. The culture medium was refreshed every two days, and the culture condition was 37 °C and 5% CO_2_. For the immunofluorescence research, the cells were fixed in the 4% paraformaldehyde at room temperature for 20 min and permeabilized with 0.1% Triton X-100 for 5 min and subsequently blocked with 1% bovine serum albumin (BSA) for 20 min at room temperature. Then, the 5 μg/mL fluorescein isothiocyanate (FITC)-Phalloidin and DAPI (4′,6-diamidino-2-phenylindole) staining were utilized to observe the cytoskeletal structure with a laser confocal microscope.

### 2.8. Antibacterial Tests In Vitro

*S. aureus* and *E. coli* were chosen to evaluate the antimicrobial performance of the coating samples. LB medium and LB agar medium were utilized to cultivate the bacterial and colony counts and printing, respectively. The antimicrobial activities of the coatings were determined by three parallel experiments for each group.

To investigate the bacterial morphology and number of coatings, the bacterial (100 μL) were cultured on the surface of the substrates in a 12-well culture plate at the initial density of 1 × 10^6^ CFU/mL. After culturing for 24 h, the 100 μL of bacteria were subsequently cultivated on the LB agar plate for another 24 h to determine the colony counts. The substrates with bacteria were fixed by 2.5% glutaraldehyde at 4 °C overnight and subsequently fixed by 1% acid solution in a fume hood for 1–2 h, dehydrated sequentially by gradient ethanol (30%, 50%, 70%, 80%, 90%, 95%, and 100% *v/v*) gradually, and critical point dried with a critical point drier (Hitachi HCP-2). Finally, the substrates were coated with gold powder for SEM observation (Hitachi SU-8010/Gemini SEM 300).

To investigate the inhibition zone of the substrates, the coating-treated substrates were placed on the LB agar medium plate inoculated with bacteria with an initial density of 5 × 10^6^ CFU/mL. After culturing for 24 h, the size of the inhibition zone around the substrate was determined for antibacterial evaluation.

### 2.9. Antibacterial Tests In Vivo

The rat osteomyelitis intramedullary nail model was used to study the anti-infection ability of the modified titanium-based rods. Wistar rats (male, 4 weeks old; about 200 ± 5 g, all animals approved by the Experimental Animal Ethics Committee of Wuhan University) were randomly divided into three groups. A total of 1.5% sodium pentobarbital (4.0 mL/kg) was injected intravenously for general anesthesia. A 1 cm longitudinal skin incision was made above the lateral femoral condyles of each animal. A custom handheld drill (1 mm) was used to drill the bone marrow cavity at the lower end of the femur intercondylar fossa with low rotation speed and saline irrigation. Then, the modified round titanium rod (Ø 1 mm × 20 mm) was inserted into the bone marrow cavity, and finally, the subcutaneous tissue and skin was sutured layer by layer with 5-0 surgical suture (Vicryl).

On the 30th day after the implantation of the reverse intramedullary nail mold for acute osteomyelitis, the knee joint cavity was opened again by surgery, and 10 μL of *S. aureus* (1 × 10^6^ CFU/mL) was injected into the bone marrow cavity with a 25 μL micro syringe (Hamilton) to develop a model of implant infection. Then, the incision was sutured, and the sutures were raised carefully. After 5 days, all 20 rats were sacrificed, the modified titanium-based needles in the bone marrow cavity were taken out, and then quickly inoculated on sterile agarose medium for 5 s, and the culture dish was placed in a bacterial incubator at a constant temperature (37 °C) wet culture for 24 h to observe the presence and size of bacterial imprints.

The removed femur was fixed in 10% formalin buffer for 48 h and then decalcified with disodium ethylenediaminetetraacetic acid solution (EDTA, 12% *v/v*) for 28 days. Hematoxylin and eosin (H&E), Gram staining, and Masson staining were performed, and the infiltration state of inflammatory cells (neutrophils, lymphocytes, and bacteria) was observed with a Leica microscope.

### 2.10. Experimental Process of Device Transformation

We transformed this multifunctional antimicrobial coating technology into commonly used implants in orthopedics, such as locking compression plates (LCP) and pedicle screws (PS). *S. aureus* and *E. coli* were utilized to culture on the surface of the modified implants to observe the bacterial morphology and to evaluate the antimicrobial performance. The bacterial elution was inoculated on the LB agar to observe the number of bacterial colonies.

### 2.11. Statistical Analysis Methods

The experimental data were all expressed in terms of mean ± standard deviation (means ± SD), and the comparison between univariate and different grouping data adopted a one-way analysis of variance. The difference was statistically significant with *p* < 0.05. Statistical analysis was performed using origin software.

## 3. Results

### 3.1. Fabrication of the PVP-I_2_ Integrated-PCL Coating

In a typical process, as shown in Figure 1A, MAO is in the mixed H_2_SO_4_/H_3_PO_4_, and H_2_O_2_ solution by anodic sparking led to the in-situ growth of porous TiO_2_ structure on titanium alloy. The resulting surface was rich in porous structure with an inner diameter varied between 1–5 μm. The cross-sectional thickness of MAO coating is about 2 μm (Appendix A). It was subsequently subjected to electrophoresis in mixed PVPI/KI solution to load the iodine in the porous structure (Appendix A). Finally, the biodegradable PCL coating was deposited by immersing the treated substrates in the polymeric solution. Additionally, the surface topography and iodine content of different substrates was determined by scanning electron microscope (SEM) and EDS mapping. Results are illustrated in Figure 1B, which indicates that iodine was incorporated in the PCL coating on the titanium alloy, and iodine content on the surface of MAO-I_2_, MAO-I_2_-PCL1, MAO-I_2_-PCL2, and MAO-I_2_-PCL3 was about 10.31 wt%, 9.02 wt%, 8.38 wt%, and 7.15 wt%, respectively.

### 3.2. Characterization of the PVP-I_2_ Integrated-PCL Coating

XPS was performed to determine the element composition on the surface of the titanium alloy. As shown in Figure 2, Ti 2p, C 1s, and O 1s peaks were detected on the both MAO and MAO-I_2_ surfaces. The Ti 2p at 458.31 eV, C 1s at 284.32 eV, and O 1s at 530.35 eV peaks corresponded to TC4 and TiO_2_ of MAO and MAO-I_2_. Additionally, the iodine signal at 617.75 eV was mainly due to the PVPI and I_2_ on the MAO-I_2_ group from electrophoresis and redox reaction. Certainly, there was no I 3d peak on the surface of MAO. According to the AFM images, the surface roughness of Ti, MAO, MAO-I_2_, MAO-I_2_-PCL1, MAO-I_2_-PCL2, and MAO-I_2_-PCL3 was 163.999 nm, 584.452 nm, 578.447 nm, 526.472 nm, 444.695 nm, and 485.828 nm, respectively (Figure 2K). The porous micro-array structure on the surface of the titanium alloy (MAO) significantly increased the surface roughness of the titanium alloy, and the subsequent PCL coating reduced the surface roughness of the MAO-I_2_.

### 3.3. Controlling Release of Iodine by the Stabilization of PCL Coating

The sustained release of antibacterial iodine is the key to the success of our antibacterial strategy. The diagram of the anodizing porous membrane and PCL coating were shown in Figure 3A. The PLC coating was generated on the surface of titanium, which can conserve iodine in nanotubes and control the release of iodine. As shown in Figure 3B, the iodine in MAO-I_2_ explosively released from the substrate surface in 3 days in the Tris buffer solution, while a very slow-release curve up to 30 days in MAO-I_2_-PCL1, MAO-I_2_-PCL2, MAO-I_2_-PCL3. Interestingly, the higher the concentration of PCL, the slower the release rate of iodine. Additionally, the results of cell viability assays and cytoskeletal staining are illustrated in Figure 3C,D, respectively. The results demonstrated that the incorporation of PVPI together with PCL did not cause detectable cytotoxicity, indicating the MAO-I_2_-PCL coating does not sacrifice the biocompatibility of the titanium alloy.

### 3.4. Antibacterial Properties of Samples In Vitro

To examine the antibacterial performance of the samples in vitro, we conducted bacterial culture experiments and dead/live staining experiments (Figure 4). In the bacteria culture experiments, both *S. aureus* and *E. coli* were directly cultured on all substrates, and a substantial bacteria reduction with membrane concavities and wizened status were detected on the surface of MAO-I_2_, MAO-I_2_-PCL1, MAO-I_2_-PCL2, and MAO-I_2_-PCL3 groups. Among them, the MAO-I_2_ group has the strongest antibacterial ability, and there are almost no bacteria on its surface. On the contrary, a lot of bacteria in Ti and MAO groups maintain their inherent elliptical or rod-shaped morphology. In addition, the live and dead staining of the two kinds of bacteria was also consistent with the above performance.

Subsequently, time-dependent antibacterial experiments in vitro were conducted on samples at different time points. As shown in Figure 5A, the Ti and MAO groups did not exhibit antibacterial properties while the MAO-I_2_ and MAO-I_2_-PLC2 groups showed two excellent circles of inhibition at 0 h after coating preparation. When these iodine-loaded substrates were left at room temperature for 30 days, their antibacterial effects changed interestingly. The MAO-I_2_ group has lost its antibacterial properties, while the MAO-I_2_-PLC2 group still has good antibacterial properties with a huge circle of inhibition. As illustrated in Figure 5B, the bacterial culture plates of *S. aureus* and *E. coli* have the same trend as the inhibition zone on the first day, fourth day, and thirtieth day. On the first day, there were a large number of bacterial colonies on MAO but no obvious colony formation on the surface of MAO-I_2_ and MAO-I_2_-PCL2 groups. From the fourth day to the thirtieth day, there are still a lot of bacteria colonies in the MAO group. However, the MAO-I_2_ group had lost its antibacterial effect with obvious bacterial colonies on the plate on the fourth day. Moreover, the colonies of the MAO-I_2_ group on the thirtieth day became more obvious. For the MAO-I_2_-PCL2 group, there is no obvious bacterial colony formation on the plate after being placed for 4 days. Even after 30 days, there are still only a few colonies on the plate.

### 3.5. Antibacterial Properties of Samples In Vivo

As shown in Figure 6A, an intramedullary nail model of the femoral was constructed in the rat. After 30 days post-operation, *S. aureus* was injected into the bone marrow cavity. On the fifth day after the injection of bacteria, the knee joint was incised to open the femoral bone marrow cavity, and the titanium rod was exposed. In the MAO and MAO-I_2_ groups, there were many infected abscesses or bacterial mucus on the surface of the knee joint and around the titanium rod. On the contrary, no obvious signs of infection were detected in the MAO-I_2_-PCL2 group, indicating that the MAO-I_2_-PCL2 group still has a strong antibacterial effect 30 days after implantation in the body (Figure 6B). Hematoxylin and eosin staining (HE) and Gram staining (Gram) were used to further evaluate the inflammatory response and residue bacteria at the infected site in the body. As shown in Figure 6C, a large number of monocytes and neutrophils (blue arrows) and *S. aureus* residues (red circles) were observed in the MAO and MAO-I_2_ groups. Similarly, no obvious bacteria and inflammatory cells were detected in the MAO-I_2_-PCL2 group, indicating that the MAO-I_2_-PCL2 group has effectively eliminated the invading bacteria. In Masson staining, there are a large number of muscle fibers in the MAO and MAO-I_2_ groups while few muscle fibers are in the MAO-I_2_-PCL2 group. Meanwhile, there are a large number of collagen fibers in these three groups.

### 3.6. Transformation Performance of Medical Devices

In order to explore the transformation performance, we transferred the whole technological process to irregular orthopedic implants. As shown in Figure 7A–C, the surface of the locking compression plates, pedicle screws, and locking screws become brunet after micro-arc oxidation and electrophoresis reaction with iodine. Then, a uniform transparent coating film can be found on the surface of devices on the MAO-I_2_-PCL2 group without obvious color change. As shown in Figure 7D, the surface topography and iodine content were demonstrated on the surface of the devices. As expected, the MAO-I_2_-PCL2 of medical devices has an excellent antibacterial performance in the bacteria culture experiments (Figure 7E) and agar plate experiments (Figure 7F).

## 4. Discussion

Implants infection, especially the late implant infection after surgery in the orthopedics field, is still a catastrophic complication, and most clinically implanted devices do not have effective antibacterial strategies [3,8]. Moreover, when bacteria colonize on the surface of the implants, the late infection, antibiotics resistance, and multiple infections make the situation worse [9,22]. Although there have been some reports of antibacterial implant research, it is difficult to tackle the late infection after orthopedic surgery due to the burst release and limited antibiotic loading of these implants [19,26]. Therefore, synergistically developing a control-released coating containing iodine on the surface of a titanium alloy may be a promising strategy to solve these challenges. In this work, we firstly demonstrated an iodine-loading underlying layer through macro-arc oxidation, followed by a PCL coating strategy, which was in order to kill sessile bacteria throughout their whole lifecycle and deal with the late bone implant-related infection.

The characterization results, surface morphology, and chemical composition of the coatings proved that the hybrid MAO-I_2_-PCLx coatings (x = 1, 2, 3) were successfully developed on the titanium alloy surface. After the micro-arc oxidation treatment, the titanium alloy gains a surface with a rich porous structure [38,39]. As the atomic force microscope showed, the surface roughness of the MAO substrate increases significantly, which creates a structural basis for the electrophoretic loading of iodine and the combination of the PCL coating in the later stage [40,41,42]. The tubular titanium dioxide structure on the surface of the titanium alloy, such as a pipe, can be loaded with iodine by electrophoresis, thereby ensuring enough content and long-term release ability of iodine [43]. Due to the unstable character of iodine, the iodine in the titanium dioxide tube will quickly sublimate and volatilize [44,45]. Thus, it is required to construct a sealable coating to block it. We creatively used PCL as this sealing coating, realized by dip-coating technology; the rough surface formed by micro-arc oxidation in the early stage (as shown by atomic force microscope) will also significantly increase the binding force between the PCL coating and the titanium alloy [46]. The stability of PCL coatings might be attributed to the thickness and density of PCL coatings and the bonding strength between the PCL coatings and the mac-oxidation surface of titanium [47]. It has been reported many times that the PCL coating has good biocompatibility [48,49,50], which makes the antibacterial coating’s iodine antibacterial properties, controlled release properties, long-lasting release properties, good biocompatibility, and stability perfectly combined.

Results of EDS and iodine release of ICP data showed that PCL-controlled coating could coordinate with iodine and control its release in various MAO-I_2_-PCLx coatings [51]. Iodine is carried by anodic oxidation, electrophoresis, and redox reaction on the underlying layer of titanium alloy, followed by a controlled release coating of PCL constructed on the outer layer, which can effectively organize the explosive release of iodine to exert a long-lasting antibacterial effect. According to the release curve of iodine, we concluded that as the concentration of PCL increased, the stability of PCL coatings increased, and the iodine release decreased. Most importantly, there is still a certain amount of iodine remaining in the MAO-I_2_-PCLx coating system after 30 days, which allows the implants to play a long-acting antibacterial and late antibacterial performance effectively. Secondly, many studies have shown that iodine is an essential trace element for the human body and plays an important role in regulating the physiological functions of normal cells [52,53]. PVPI is a widely used disinfectant in clinical practice [54,55]. It has been verified for a long time that it can be applied to the human body as an antibacterial agent and has good biocompatibility [56]. The CCK-8 and cell cytoskeleton staining have determined that the iodine in the MAO-I_2_-PCLx coating has great cell biocompatibility. Therefore, loading iodine on the surface of titanium alloy through PCL-controlled coating is an excellent method for clinical transformation and application.

The in vitro antimicrobial experiments proved that MAO-I_2_-PCLx coating possessed superior antimicrobial performance against sessile *S. aureus* and *E. coli* over other groups. Since PCL has not been reported to have significant antibacterial properties [57,58], we believe that the underlying iodine layer by macro-arc oxidation and electrophoresis is responsible for its superior antimicrobial ability [59,60]. Especially, the antibacterial results in vitro revealed that the controlled PCL coating could release the iodine slowly and steadily. It should be noted that the iodine release curve of MAO-I_2_-PCL2 coating could maintain for approximately 30 days. The high peak concentration of iodine could eliminate the late infection and the sessile bacteria in one month. Therefore, we believe that the MAO-I_2_-PCLx coatings can combine the superior late antibacterial activity and good cell biocompatibility synergistically in order to kill stubborn bacteria and protect the normal surrounding tissue of the body.

Meanwhile, we evaluated the antibacterial activity of MAO-I_2_-PCLx coatings of titanium alloy rods using an implant-related osteomyelitis rat model. The results of the in vivo tests revealed that the iodine-loading PCL coating exhibited excellent late antibacterial ability. The antimicrobial process and mechanism of iodine-loading PCL coating in vivo are much more complicated than the experiments in vitro, because of the body’s physiological environment and degradation of coating in the body [61]. It has been demonstrated that residual iodine of the coatings has strong antibacterial capacities by inducing the disintegrating of a lot of bacteria in vivo even after 30 days. We also found that the inflammatory response and residual bacteria of the MAO-I_2_-PCLx were much lower than the other groups from the results of the microbiological and histological analysis. Therefore, we speculate that PCL-controlled coating exerts an outstanding control release ability to maintain the iodine content in the coating system to eliminate the late sessile bacteria.

Because titanium alloy implants were very widely used in orthopedics, this surface modification strategy of titanium alloy has great clinical application value [62,63,64]. However, due to the extremely unstable characteristics of iodine and the surface hardness of titanium alloy, it is very difficult for the combination of iodine and titanium alloy to exert long-lasting antibacterial properties [65]. In order to overcome this problem, we intelligently use the surface-treated method of anodization to gain the structure with a high surface area, which could load a large amount of iodine. Ultimately, a sophisticated multilayered system with an underlying layer of iodine and an upper layer of LCP was implemented, which would be most efficacious and convenient for the device to continuously and slowly release antibacterial iodine from coatings, which effectively achieves long-term antibacterial properties. Moreover, the coating technology was successfully translated into clinically orthopedic implants, likely pedicle screws and locking compression plates. Collectively, it is clearly demonstrated that the MAO-I_2_-PCLx coatings could be translated into orthopedic implants and further exert great antibacterial capacity to eliminate the late infection in the orthopedic fields.

Nevertheless, this MAO-I_2_-PCLx coating strategy also has some shortcomings. Because of the limited bonding strength between the PCL coating and the titanium alloy surface, the PCL coating was easily damaged under huge external force, which may thereby affect its controlled release effect. Therefore, in our following research, we will focus on this point to further improve the bonding strength of the titanium alloy and the PCL coating. At the same time, further extending the period of iodine release and the effective antibacterial content are our research priorities.

## 5. Conclusions

In this work, an MAO-I_2_-PCLx coating strategy was developed for the anodization followed by iodine-loading and PCL-coating that could be successfully constructed on the surface of the surgical prosthesis. It was demonstrated that the steady ability of iodine release was gradually increased by the PCL-controlled coatings. Results of cell and bacterial assays in vitro and in vivo revealed that the MAO-I_2_-PCLx coating has superior late antibacterial properties and great cell biocompatibility. LCP coating delayed release of iodine from the underlying coating could exert a great antibacterial effect and promote the proliferation of BMSCs. Most importantly, the underlying iodine release makes it possible for the system to eliminate the late infection. Therefore, with the incorporation of biologically essential iodine under the LCP overlayer for controlled release, the gradient functional modification systems are suitable for different titanium alloy fields for which the risk of infection has not been completely addressed in orthopedics.

## Figures and Tables

**Figure 1 jfb-13-00195-f001:**
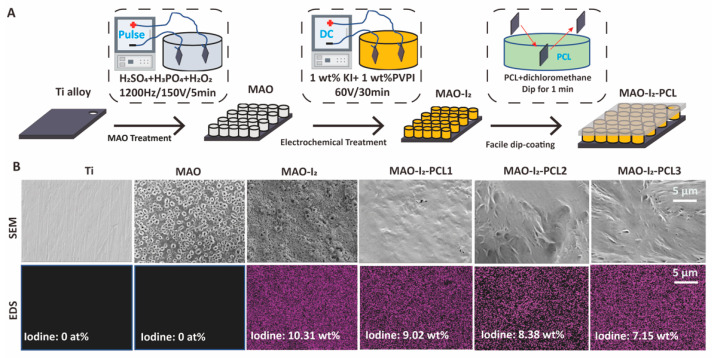
Schematic route and surface characteristics of the MAO-I_2_-PCLx coatings (x = 1, 2, 3). The illustrative diagrams of the preparation process (**A**), SEM morphology (**B**), and EDS mapping (**B**) of iodine-carrying PCL-controlled release coating.

**Figure 2 jfb-13-00195-f002:**
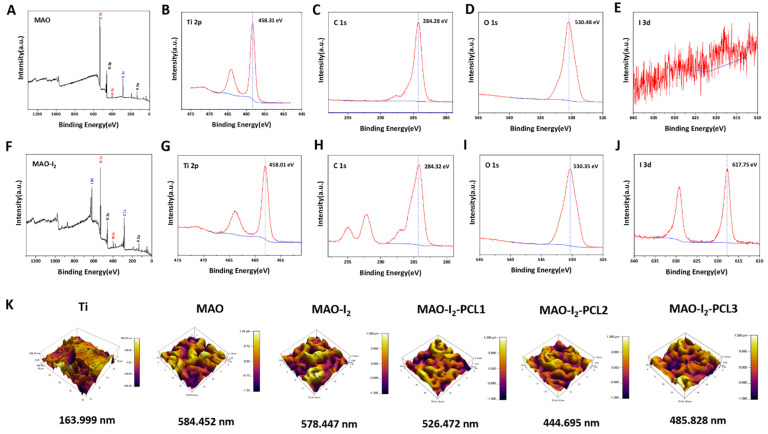
Chemical composition and surface roughness of the coatings: (**A**) XPS results about the survey spectrum of MAO coating; (**B**–**E**) the corresponding core-level spectra for Ti 2p, C 1s, O 1s of MAO coating; (**F**) XPS results about the survey spectrum of MAO-I_2_ coating; (**G**–**J**) the corresponding core-level spectra for Ti 2p, C 1s, O 1s of MAO-I_2_ coating; (**K**) the AFM images of the titanium and MAO-I_2_-PCLx coatings (x = 1, 2, 3).

**Figure 3 jfb-13-00195-f003:**
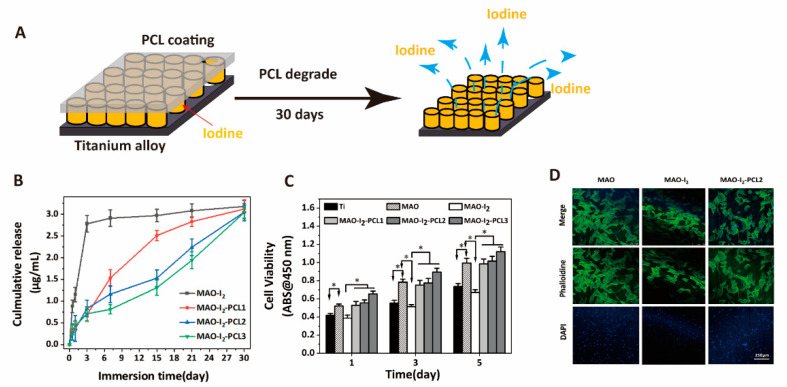
The time-dependent iodine release curve and iodine content determination of the coatings: (**A**) schematic diagram for the iodine release of coating; (**B**) ICP determination and analysis of the cumulative release content in Tris buffer at different time points; (**C**) cell biocompatibility test of different coatings; and (**D**) cytoskeleton staining after 3 days of BMSC growth on the coatings. * denotes *p* < 0.05.

**Figure 4 jfb-13-00195-f004:**
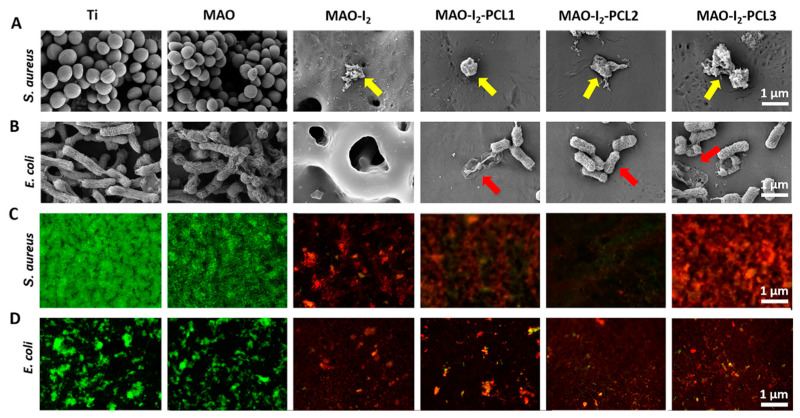
The antibacterial activities of the coatings through a three-day exposure to *S. aureus* and *E. coli*: (**A**,**B**) the SEM micrographs of cultured bacteria on the surface of coatings (yellow arrows represent the depressed *S. aureus* and red arrows represent the depressed *E. coli*); (**C**,**D**) live/dead confocal images showing the live or dead bacteria on the coatings (green represents the live bacteria and red represents the dead bacteria).

**Figure 5 jfb-13-00195-f005:**
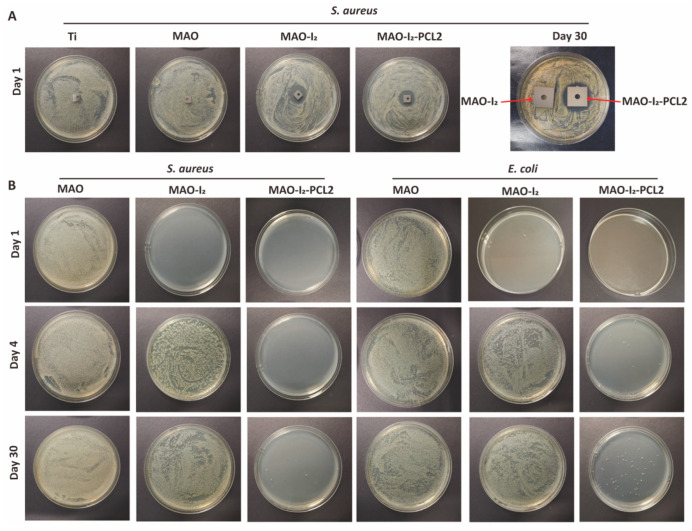
Antibacterial experiments of corresponding substrates at different time points in vitro: (**A**) inhibition zone experiment of different samples of *S. aureus*; (**B**) colonies of *S. aureus* and *E. coli* were cultivated on the corresponding surfaces of samples at different time points.

**Figure 6 jfb-13-00195-f006:**
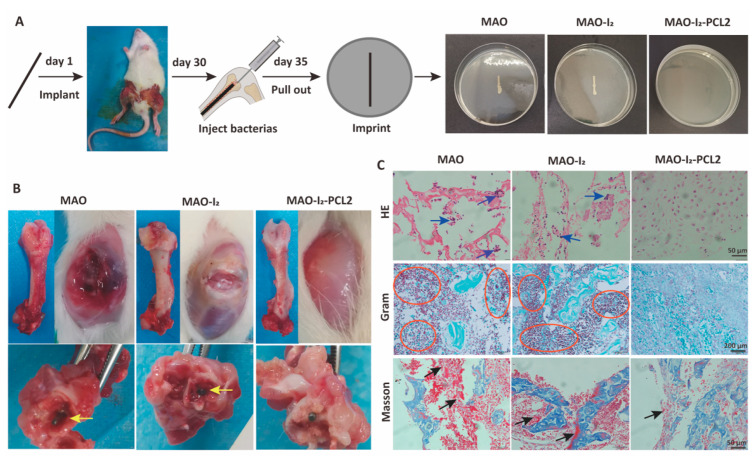
Antibacterial effect of the corresponding substrates in vivo: (**A**) scheme of implantation process of implanting a titanium rod in the femur; (**B**) the infected rat femur images 7 days after the implantation of the titanium rod; (**C**) H&E, Gram, and Masson staining images in the regions of femur osteoepiphysis with different samples implantation, respectively; blue arrows represent lymphocytes and neutrophils infiltration, red circles represent inflammatory cells and bacteria, and black arrows represent collagenous fibers.

**Figure 7 jfb-13-00195-f007:**
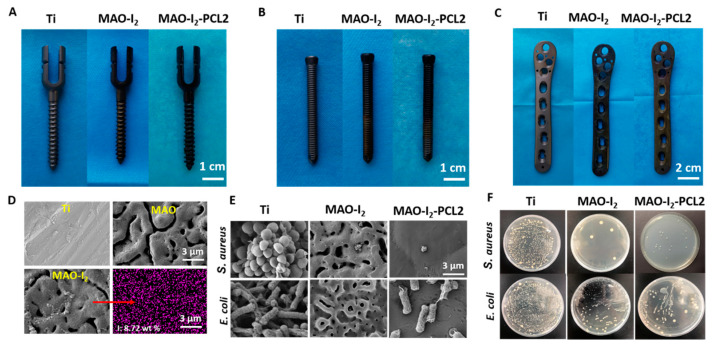
Transformation effect of orthopedic implants with the antibacterial performance: (**A**–**C**) general observation of pedicle screws (PS), locking screws (LS), and locking compression plate (LCP); (**D**) the surface morphology and EDS mapping of iodine element; (**E**) SEM morphology of *S. aureus* and *E. coli* culturing on the surface of the LCPs; (**F**) spread plate images of *S. aureus* and *E. coli* seeding on the surface of the LCPs.

## Data Availability

The data that support the findings of this work are available upon reasonable request from the authors.

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
