# Peer review of "Remote Eradication of Bacteria on Orthopedic Implants via Delayed Delivery of Polycaprolactone Stabilized Polyvinylpyrrolidone Iodine"

_jfb, 2022, doi:10.3390/jfb13040195_

Round 1
Reviewer 1 Report
I have reviewed the manuscript jfb-1956474 entitled "Remote Eradication of Bacteria on Orthopedic Implants via Delayed Delivery of Polycaprolactone Stabilized Polyvinylpyrrolidone Iodine" proposed by Yikai Wang et al. for publication in the Journal of Functional Biomaterials. The results are interesting. However, some parts of the manuscript should be improved before it is considered to be acceptable. In my opinion, the following points have to be considered:
1. The study is quite comprehensive and has achieved very good results. Authors should write the abstract section in more detail.
2. The authors did not explain the novelty and significance of their work in the introduction section. Moreover, this section is not cohesive. Indeed, this section is intended to "convey the core findings of the paper", i.e. reflect the best novelty of this paper in a concise form. The authors shall show the best novelty of the work, such as how your research advances the state-of-the-art of the topic/area, and /or how much better is your work compared with peer researchers on the same or similar topics. At the end of this section, the main objective of this study must be mentioned.
3. The experimental section is poorly written. What type of power supply has been used for the MAO process? If the homemade power supply was used how one could ensure reproducibility of the results? Many essential details of surface characterization and evaluation are missing. The description is virtually reduced to listing the methods and instruments used, rather than details of procedures and subsequent analysis.
4. How MAO parameters (voltage 150V, 141 frequency 1200HZ, duty cycle 10%, and time 5 min) were determined, and how was the thickness of the MAO coatings?
5. The background and state-of-art of MAO should be improved and updated since there are many publications recently. Indeed, more literature should be cited in the introduction about anti-bacterial agents’ incorporation in the MAO electrolytes for Ti-based implants, for example, Materials Chemistry and Physics 276 (2022): 125436.
6. On page 2, line 54, the sentence “over the past three decades, multiple drug-device combination system has been attempted to prevent the infection such as antibiotic-loaded implants ….” needs the following reference: Materials Science and Engineering: C 71 (2017): 1241-1252.
7. For the fabrication of MAO coating, the breakdown voltage and breakdown time should be given in the first paragraph of section 3.1 or in section 2.3.2.
8. What is the interfacial quality between coatings and substrate (adhesion strength)? In this work, there are no cross-sectional SEM images. The cross-sectional SEM image is suggested to be supplemented.
9. On page 14, line 441, the sentence “As the atomic force microscope showed, the surface roughness of the MAO substrate increases significantly, …” needs the following reference: Journal of Functional Biomaterials 13.2 (2022): 50
10. There are some formatting mistakes in the references section, I suggest the authors check and correct them. For example, there are incomplete references or with erroneous data, others with typos in the journal name or chemical formulae in the title. For example, this issue can be seen in ref. no. 37.
Author Response
October 8, 2022
Dear reviewer,
We are very thankful to you for your thorough review of our manuscript. We have carefully addressed their excellent questions point by point and revised the manuscript accordingly. Please find below our responses (in blue letters after a “>>” symbol) to each of the reviewers' critiques. We feel that the manuscript has been significantly improved and hope it is now acceptable by Journal of Functional Biomaterials.
Sincerely yours,
Yijun Ren, Ph. D.
Department of Orthopedics, Renmin Hospital of Wuhan University, 238# Jiefang Road, Wuhan, Hubei Province 430060, P.R. China
Tel: (+86) 027-8804-1911-83380.
E-mail: renyj6969@whu.edu.cn
Zhaoming Ye, Ph.D.
Department of Orthopedics, the Second Affiliated Hospital,
Zhejiang University School of Medicine,
88# Jiefang Road, Hangzhou, Zhejiang 310009, P.R. China
Tel: (+86) 571-8778 3777
E-mail: yezhaoming@zju.edu.cn

Reviewer 2 Report
Dear Authors,
Please see the attachment.

Author Response

(The authors gave the same response as above.)

Round 2
Reviewer 1 Report
Accept in the present form! The authors were successful in answering the reviewer's comments.